# Hypoglycemia prevention practice and its associated factors among diabetes patients at university teaching hospital in Ethiopia: Cross-sectional study

**Esileman Abdela Muche** **\*, Banchamlak Teferi Mekonen**

Department of Clinical Pharmacy, School of Pharmacy, College of Medicine and Health Sciences, University of Gondar, Gondar, Ethiopia

\* sulamanabdela@gmail.com

**Data Availability Statement:** All relevant data are within the manuscript and its Supporting Information files.

## Abstract

### Introduction

Hypoglycemia is a true medical emergency, which needs prompt recognition and treatment to prevent organ damage and mortality. Knowledge about the prevention of hypoglycemia is an important step to self-care practice because informed people are more likely to have a better hypoglycemia prevention practice. The aim of this study was to explore hypoglycemia prevention practice and its associated factors among diabetes patients at a university teaching hospital in Ethiopia.

### Method

A cross-sectional study was carried out on a total of 422 systematically selected diabetic patients at the University of Gondar Referral and Teaching Hospital. Data were collected using a pre-tested, structured, and interviewer-administered questionnaire. The collected data was analyzed by SPSS version 20 and associated variables were measured using binary logistic regression and within 95% confidence interval. A p-value <0.05 was considered as statistically significant.

### Result

From the total of 422 diabetic patients, 61.6% were males, 70.1% of them were urban dwellers, 37.9% of them were unable to write and read, and 70.6% of the participants were taking insulin. The majority of respondents had good knowledge of (77.5%) and practice of (93.1%) hypoglycemia prevention. Only good participant knowledge about hypoglycemia prevention was strongly associated with the practice of its prevention (AOR: 2.87 (1.2–6.8), p = 0.01).

### Conclusion and recommendation

Even though diabetic patients with good knowledge of hypoglycemia and its prevention was strongly associated with good prevention practice, there exists a gap in knowledge of

**Funding:** The author(s) received no specific funding for this work.

**Competing interests:** The authors have declared that no competing interests exist.

**Abbreviations:** AOR, Adjusted Odd Ratio; BMI, Body Mass Index; COR, Crude Odd Ratio; CI, Confidence Interval; DM, Diabetes Mellitus; OPD, Out-Patient Department; SD, Standard Deviation; SMBG, Self- Monitoring Of Blood Glucose; UoGCSH, University of Gondar Comprehensive and Specialized Hospital.

hypoglycemia prevention. Hence, we recommend counseling be offered to patients regarding hypoglycemia during their visit to the diabetic clinic. Counseling points such as common clinical symptoms, its negative consequence, as well as remedial options are essential elements for the improvement of their practice on its prevention.

# Introduction

Diabetic Mellitus is a non-communicable heterogeneous group of metabolic disorders with elevated blood glucose and abnormally shifted carbohydrate, fat and protein metabolism resulting from defects in insulin secretion and/or insulin action [1]. Both acute and chronic complications are responsible for the death and hospitalization associated with diabetes. Hypoglycemia, which is defined as "an abnormally low plasma glucose concentration ($<$70 mg/dl) that exposes the subject to potential harm", is one of the acute complications of diabetes mellitus [2, 3].

A systemic review and meta-analysis conducted among type II DM patients reported the prevalence of mild/moderate and severe hypoglycemia to be 45%, 6% respectively [4]. Another multi-center study that aimed to assess rates and predictors of hypoglycemia reported that 83.0% of patients with type I DM and 46.5% of patients with type II DM experienced hypoglycemia [5].

Hypoglycemia poses a significant economic burden on the health care system through frequent emergency room visits, ambulance utilization, and hospitalizations costs. An estimated 2–4% of people with type 1 diabetes mellitus die from this complication of DM each year [6–11].

Insulin therapy, insulin secretagogues, skipping a meal, doing physical exercise without taking food, a history of severe hypoglycemia, alcoholic beverages, renal insufficiency, coronary artery disease, and infections are the most common reasons for the recurrent episodes of hypoglycemia [12–15].

The spectrum of symptoms depends on the duration and severity of hypoglycemia and vary from autonomic activation to behavioral changes to altered cognitive function. Checking blood glucose levels is the only way to know whether a person is experiencing low blood glucose. The short and long-term complications include neurologic damage, trauma, cardiovascular events, and death [6, 12].

Hypoglycemia treatment requires the ingestion of glucose or carbohydrate-containing foods. Pure glucose is the preferred treatment, but any other form of carbohydrate which contains glucose will help to raise blood glucose. Glucagon is indicated for the treatment of hypoglycemia in people unable or unwilling to consume carbohydrates by mouth [14, 16]. However, early detection and prevention are preferred to its treatment to avoid severe negative health sequela and economic burden [17].

Effective approaches known to reduce the risk of hypoglycemia include patient education along with self-monitoring of blood glucose (SMBG), dietary modifications and regular exercise, medication adjustment, careful glucose monitoring by the patient, and conscientious follow up by the clinician [18, 19]. Moreover, knowledge about symptoms of hypoglycemia is an important step to self-care practice, because informed people are more likely to have better practice [20]. Having good knowledge about hypoglycemia is positively associated with good hypoglycemia prevention practice [21].

Several cross-sectional studies on knowledge of hypoglycemia symptoms reported as being poor or good. Research findings showed that 64.4% of diabetic patients had good knowledge

of hypoglycemia [22], while some studies done in a rural population indicated that 63.33% of diabetic patients had inadequate knowledge [23]. However, some other studies showed that more than half of the study participants had knowledge about symptoms associated with hypoglycemia [24, 25]. In other study conducted in Ethiopia, 63.2% of participants had good hypoglycemia prevention practice [21]. They recommended educating the patient as one strategy to have a better practice on the prevention of hypoglycemia.

Even though there are many research works reported on self-care practice and knowledge about hypoglycemia, there are no studies done on knowledge and practice regarding hypoglycemia prevention among DM patients who are on regular follow up at the University of Gondar Referral and Teaching Hospital. Therefore, the current study aimed at assessing the status of knowledge and practices among diabetic patients towards the prevention of hypoglycemia and its complication at the University of Gondar Referral and Teaching Hospital (UoGRTH).

## Methods

### Study area

The study was conducted at the chronic outpatient department (OPD) of the University of Gondar Referral and Teaching Hospital (UoGRTH), which is located in Gondar town, Amhara national regional state, which is 750 km far from Addis Ababa, North-west Ethiopia. The hospital serves for more than 7 million people, who are found in central, north, and west Gondar zones and the surrounding zones and woredas. It has 680 beds and 21 wards for inpatients, emergency, and outpatient department services.

**Study design and period.** A hospital-based cross-sectional study was conducted from February-March 2019.

**Population.** *Source population*. All diabetic patients who were attending chronic OPD at UoGRTH.

*Study population*. All diabetic patients who were attending at UoGRTH chronic OPD during the study period.

**Inclusion and exclusion criteria.** *Inclusion criteria*. All adult diabetic patients who were attending chronic OPD.

*Exclusion criteria*. Patients who were seriously ill and unable to communicate, patients who were unwilling to participate, and women with gestational diabetes.

**Sample size & sampling methods.** A single population proportion formula was used to calculate sample size, n = $z^2$p (1-p)/$d^2$, by considering P = 50% precession (d), 5% marginal error and 95% confidence interval. After adding a 10% contingency on the calculated 384 patients, the final sample size became 422. The daily average number of ambulatory diabetic patients visiting the outpatient department was estimated to be between 110 and 130. Systematic random sampling was done with a sampling interval of 5. Every fifth patient coming to the OPD was selected by starting from a random number.

**Data collection tools.** A structured questionnaire was adapted from different literatures [6, 21, 23, 26]. It consists of data regarding knowledge and practices related to the prevention of hypoglycemia, socio-demographic variables, and clinical characteristics. The knowledge assessment questionnaire has 10 questions with a maximum of 12 points which was calculated by giving one for each correct response and zero for each wrong response, except for question number three which has two correct answers; so, it has two points. The practice part of the questionnaire has 15 questions with a maximum score of 17. It was calculated by giving one for each correct response and zero for each wrong response except the first question which has two points.

**Study variables.** *Dependent variables*. Hypoglycemia prevention practice

*Independent variables*. Socio-demographic variables (age, sex, income, marital status, education, religion, and occupation). Clinical characteristics-related variables (body mass index (BMI), Types of DM, duration of treatment, type of medication used, frequency of taking medication, history of hypoglycemia, and co-morbidity).

## Operational definitions

- Good Knowledge: a score of > 6 on the knowledge assessment questions.

- Poor Knowledge: a score of < 6 on the knowledge assessment questions.

- Good Practice: a score of >8.5 on the practice assessment questions.

- Poor practice: a score of < 8.5 on the practice assessment questions.

**Data collection procedures.**   A structured questionnaire was developed in English and translated to Amharic, which was the language spoken by the study subjects. It was back-translated to English by another person to check for consistency. Three pharmacy technicians and one clinical pharmacist were recruited for data collection and supervision, respectively. The questionnaire was pretested on 21 diabetic patients in the same set up who were not included in the main study. The original questionnaire was modified for clarity, sensitiveness, and completeness based on the pre-test data.

**Data processing and analysis.**   Data analysis was done by using International Business Machines Corporation, Statistical Package for the Social Sciences (IBM SPSS) version 20). Descriptive analysis was done. Frequency distribution and percentage were used to describe results on the knowledge and practice of hypoglycemia prevention. Associated factors were identified using binary logistic regression analysis. Variables with p-value of less than 0.3 from bivariate analysis were taken to multivariate logistic regression analysis and a p-value < 0.05 was considered as statistically significant.

**Ethical consideration.**   The study was conducted after ethical approval of the proposal from the University of Gondar, School of Pharmacy ethics committee (Ref.No:SOP318/2011). A letter of cooperation was obtained from the chief medical director. Since there were so many illiterate participants and finding a witness for every illiterate individual was difficult, we took verbal informed consent from each study participant. Data was taken anonymously and kept confidential throughout the study period.

## Results

### Sociodemographic variables and clinical characteristics of study subjects'

A total of 422 DM patients participated in this study. The mean age of respondents was 42.23 (SD±16) years. More than half (61.6%) of them were male. Two-hundred and seventy (64%) participants were married. The majority (85.1%) of the participants were Orthodox Christian followers. Two-hundred and ninety-six (70.1%) of them were urban residents. Regarding their education status, 38.6% were illiterate and the rest completed primary school and above. Two-third (67.3%) of the study subjects had a normal BMI. Two-hundred and twenty-eight (54%) of the study subjects were type II diabetes mellitus patients and 17% of participants were comorbid with hypertension. Approximately 71% of the participant experienced insulin injection therapy and the rest were on oral hypoglycemic agents. In addition, 46.4% of the participant took their medications for more than five years (Tables 1 and 2).

**Table 1. Socio demographic characteristic of study participants, UoGRTH, North-West Ethiopia, 2019 (N = 422).**

| Variable | Classification | Frequency (%) |
|---|---|---|
| Age | 18–34 | 154 (36.5%) |
| | 35–64 | 220 (52.1%) |
| | > 65 | 48 (11.4%) |
| Sex | Male | 260 (61.6%) |
| | Female | 162 (38.4%) |
| Residence | Urban | 296 (70.1%) |
| | Rural | 126 (29.9%) |
| Religion | Orthodox | 359 (85.1%) |
| | Muslim | 56 (13.3%) |
| | Protestant | 7 (1.7%) |
| Marital status | Unmarried | 97 (23%) |
| | Married | 270 (64%) |
| | Divorced | 21 (5%) |
| | Widowed | 34 (8.1%) |
| Educational status | Unable to write and read | 160 (37.9%) |
| | Primary education | 112 (26.5%) |
| | Secondary education | 87 (20.6%) |
| | College and above | 63 (14.9%) |
| Occupation | Unemployed | 102 (24.2%) |
| | Private | 100 (23.7%) |
| | Government | 85 (20.1%) |
| | Student | 35 (8.3%) |
| | House wife | 41 (9.7%) |
| | Farmer | 59 (14%) |
| Income per month | <15USD* | 263 (62.3%) |
| | 15–30 USD | 49 (11.6%) |
| | 30-45USD | 24 (5.7%) |
| | >45USD | 86 (20.4%) |

*USD—United states dollar

## Knowledge about hypoglycemia prevention

Of all the participants, 327 of them (77.5%) had good knowledge about hypoglycemia prevention. Most (82.2%) of the participants knew about the blood glucose level below which it is termed hypoglycemia. The majority (90.5%) of the study subjects correctly answered questions with regard to the causes of hypoglycemia. Three-hundred fifty-four (83.9%) of the study subjects knew the symptoms of hypoglycemia and 295 (69.9%) of the participants knew about ways to prevent hypoglycemia (Table 3).

## Practice experience on hypoglycemia prevention

Among the respondents, 393 (93.1%) had good practice in hypoglycemia prevention. For example, the majority (87.2%) of participants responded that they carry simple sugar while traveling. Similarly, 74 (17.5%) participants reported that they monitor their blood glucose level at home, and few (8.8%) participants practiced re-testing their blood glucose after managing a hypoglycemia incident. Almost all (95.5%) of participants were adherent towards the regular appointments, and the majority (93.3%) of participants reported hypoglycemia episodes to their physician (See Table 4).

**Table 2. Clinical characteristics of the study participants, UoGRTH, Northwest-Ethiopia, 2019 (N = 422).**

| Variable | Classification | Frequency |
|---|---|---|
| BMI | < 18.5 | 51 (12.1%) |
|  | 18.5–24.5 | 284 (67.3%) |
|  | 25–29.9 | 79 (18.7%) |
|  | >30 | 8 (1.9%) |
| Types of diabetes | Type I | 194 (46%) |
|  | Type II | 228 (54%) |
| Duration with illness (DM) in year | 1–2 | 102 (24.2%) |
|  | 3–5 | 124 (29.4%) |
|  | >5 | 196 (46.4%) |
| Types of treatment | Insulin | 298 (70.6%) |
|  | Metformin | 74 (17.5%) |
|  | Metformin + Glibenclamide | 34 (8.1%) |
|  | Insulin + metformin | 16 (3.8%) |
| Frequency of taking medication | Once a day | 30 (7.1%) |
|  | Twice a day | 392 (92.9%) |
| History of hypoglycemia in the last month | yes | 16 (3.8%) |
|  | No | 406 (96.2%) |
| Co-morbid condition | Hypertension | 75 (17.8%) |
|  | Dyslipidemia | 15 (3.6%) |
|  | Hypertension and dyslipidemia | 23 (5.5%) |
|  | Heart failure | 2 (0.5%) |
|  | Other* | 8 (1.9%) |
|  | None | 299 (70.9%) |

* Epilepsy, peptic ulcer disease, hyperthyroidism, stroke, constipation, endocarditis, dermatitis, osteoarthritis.

## Factors associated with practice on hypoglycemia prevention

Bivariate logistic regression analysis showed that gender (P = 0.08), educational status (P = 0.084), occupation (P = 0.013), duration of the illness (P = 0.082), the type of treatment given (P = 0.046), and the level of knowledge about prevention of hypoglycemia (p = 0.001) were factors possibly associated with the practice of hypoglycemia prevention. However, only

**Table 3. Knowledge regarding hypoglycemia prevention among study subjects, UoGRTH, North-west Ethiopia, 2019 (N = 422).**

| Variable | Good knowledge response N (%) | Poor knowledge response N (%) |
|---|---|---|
| low blood glucose level (Hypoglycemia) | 347(82.2) | 75 (17.8) |
| Normal blood glucose level | 247 (58.5) | 175 (41.5) |
| Main cause of hypoglycemia | 382 (90.5) | 40 (9.5) |
| Risk factor for hypoglycemia | 243 (57.6) | 179 (42.4) |
| Symptoms of hypoglycemia | 354 (83.9) | 68 (16.1) |
| Symptoms of night time hypoglycemia | 156 (37) | 266 (63) |
| Complication of hypoglycemia | 390 (92.4) | 32 (7.6) |
| Ways to Prevent hypoglycemia | 295 (69.9) | 127 (30.1) |
| Prevent night time hypoglycemia | 211 (50) | 211 (50) |
| Prevent repeated hypoglycemia | 193 (45.7) | 229 (54.3) |
| Total Knowledge assessment | 327 (77.5) Good Knowledge | 95 (22.5) poor knowledge |

**Table 4. Practice regarding hypoglycemia prevention among ambulatory diabetic patients, UoGRTH, North-west- Ethiopia, 2019 (N = 422).**

| Variables | Good practice response N (%) | Poor practice response N (%) |
|---|---|---|
| Have table sugar while travelling | 368 (87.2) | 42 (10) |
| Self-management of hypoglycemia | 344 (81.5) | 78 (18.5) |
| When did you experience hypoglycemia | 336 (79.6) | 86 (20.4) |
| Safe exercise to avoid hypoglycemia | 390 (92.9) | 30 (7.1) |
| Duration of exercise | 278 (65.9) | 144 (34.1) |
| Effect of weight lifting in a hypoglycemic patient | 261(61.8) | 161 (38.2) |
| Self-blood glucose monitoring at home | 74 (17.5) | 348 (82.5) |
| Measure blood glucose when you think hypoglycemic | 187 (44.3) | 235 (55.7) |
| Retest blood glucose after treatment of hypoglycemia | 37 (8.8) | 385 (91.2) |
| Taking snacks | 137 (32.5) | 285 (67.5) |
| Irregular carbohydrate diet | 62 (14.7) | 360 (85.3) |
| Coming in regular appointment | 403 (95.5) | 19 (4.5) |
| Adjust medication based on blood glucose level | 243 (57.6) | 179 (42.4) |
| Report hypoglycemia episodes to a physician | 394 (93.3) | 28 (6.6) |
| Total practice assessment | 393 (93.1) Good | 29 (6.9) Poor |

the level of knowledge of hypoglycemia prevention showed statistically significant association (p = 0.01; AOR: 2.87 95%CI (1.2–6.8) on multivariate regression (See Table 5).

## Discussion

Hypoglycemia can cause serious morbidity and even death if it is severe and prolonged. Due to the deprivation of glucose in the central nervous system, the primary clinical symptoms are neuroglycopenic presentations such as confusion, fatigue, and loss of consciousness. The purpose of this study was to assess the knowledge and practice of hypoglycemic prevention of patients with diabetes mellitus. In this study, we found that more than two-thirds (72.7%) of the participants had good knowledge about hypoglycemia prevention. This finding is higher than the study conducted in South Gondar, Ethiopia (25.5%). This may be due to the fact that there is an increasing public awareness about their health status due to civilization and an increase in media coverage. The other possible reason might be differences in study participants and the study setup. Our study is conducted in a comprehensive and specialized hospital which is a university teaching hospital with a better quality of care [26]. Our current finding is slightly higher than the study conducted in South Africa (66.1%). This difference may be due to differences in patient demographics and possibly in time [25].

The knowledge of most of the study participants (82.2%) regarding low blood glucose levels (hypoglycemia) was good. This result is higher than the report from Ludhiana, India (60%) [27]. Less than half of the participants in our study (42.4%) were unaware of the risk factors for hypoglycemia. This is comparable with one study from India (43.4%) but slightly higher than another report from the same country (32%) [5, 28].

With respect to the knowledge of hypoglycemia symptoms, the majority of the current study subjects (83.9%) had good knowledge. It was higher when compared to the report from KwaZulu-Natal, South Africa (66%), and Karnataka, India (65%) [25, 28]. These differences may be due to differences in participants' demographics. There were more literate participants in our study (62%) than the Indian study (48%).

**Table 5. Factors associated with hypoglycemia prevention practice among study subjects, UoGRTH, North-west- Ethiopia, 2019 (N = 422).**

| Variable | Practice regarding prevention of hypoglycemia | | P value | COR | AOR (95% CI) | P value |
|---|---|---|---|---|---|---|
| | Good (393) N (%) | Poor (29) N (%) | | | | |
| Age | | | 0.55 | | | |
| 18–34 | 145 (36.9) | 9 (31) | 0.28 | 1.00 | | |
| 35–64 | 205 (52.2) | 15 (51.7) | 0.39 | 0.533 | | |
| >65 | 43 (10.9) | 5 (17.2) | 0.00 | 0.629 | | |
| Sex | | | | | | |
| Male | 249 (63.4) | 11 (37.9) | 0.008 | 0.353 | 2.39 | 0.10 |
| Female | 144 (36.6) | 18 (62.1) | | | | 9 |
| Residence | | | | | | |
| Urban | 277 (70.5) | 19 (65.5) | 0.573 | 0.795 | | |
| Rural | 116 (29.5) | 10 (34.5) | | | | |
| Marital status | | | 0.396 | | | |
| Unmarried | 94 (23.9) | 3 (10.3) | 0.187 | 1.00 | | |
| Married | 248 (63.1) | 22 (75.9) | 0.892 | 0.329 | | |
| Divorce | 20 (5.1) | 1 (3.4) | 0.578 | 0.916 | | |
| Widowed | 31 (7.9) | 3 (10.3) | 0.000 | 0.516 | | |
| Educational status | | | 0.084 | | | |
| Unable to write and read | 143 (36.4) | 17 (58.6) | 0.075 | 1.00 | 2.53 | 0.182 |
| Primary education | 107 (27.2) | 5 (17.2) | 0.033 | 0.393 | 6.80 | 0.061 |
| Secondary education | 85 (21.6) | 2 (6.9) | 0.545 | 0.197 | 1.83 | 0.539 |
| Graduate | 58 (14.8) | 5 (17.2) | 0.000 | 0.725 | | |
| Occupation | | | 0.013 | | 2.59 | 0.25 |
| Unemployed | 98 (24.9) | 4 (13.8) | 0.426 | 1.00 | 1.27 | 0.77 |
| Private employed | 96 (24.4) | 4 (13.8) | 0.443 | 0.561 | 0.55 | 0.60 |
| Government employed | 79 (20.1) | 6 (20.7) | 0.948 | 0.572 | 0.59 | 0.65 |
| Student | 33 (8.4) | 2 (6.9) | 0.838 | 1.044 | 0.57 | 0.51 |
| House wife | 32 (8.1) | 9 (31) | 0.034 | 0.833 | | |
| Farmer | 55 (14) | 4 (13.8) | 0.000 | 3.867 | | |
| Income | | | 0.871 | | | |
| <15USD | 244 (62.1) | 19 (65.5) | 0.938 | 1.00 | | |
| 15-30USD | 47(12) | 2 (6.9) | 0.498 | 1.03 | | |
| 30-45USD | 22 (5.6) | 2 (6.9) | 0.821 | 0.56 | | |
| >45USD | 80 (20.4) | 6 (20.7) | 0.000 | 1.21 | | |
| BMI | | | | | | |
| <18.5 | 40 (12.2) | 11 (11.6) | 0.844 | | | |
| 18.5–24.9 | 220 (67.3) | 64 (67.4) | | | | |
| 25–29.9 | 62 (19) | 17 (17.9) | | | | |
| >30 | 5 (1.5) | 3 (3.2) | | | | |
| Type of DM | | | | | | |
| Type I | 183 (46.6) | 11 (37.9) | 0.370 | 0.70 | | |
| Type II | 210 (53.4) | 18 (62.1) | | | | |
| Diabetic duration in year | | | 0.082 | | | |
| 1–2 | 92 (23.4) | 10 (34.5) | 0.634 | 1.00 | 0.40 | 0.07 |
| 3–5 | 121 (30.8) | 3 (10.3) | 0.046 | 1.22 | 3.70 | 0.05 |
| >5 | 180 (45.8) | 16 (55.2) | 0.00 | 0.27 | | |
| Type of treatment | | | 0.046 | | | |
| Insulin | 279 (71) | 19 (65.5) | 0.349 | 1.00 | 1.77 | 0.51 |

(*Continued*)

**Table 5.** (Continued)

| Variable | Practice regarding prevention of hypoglycemia | | P value | COR | AOR (95% CI) | P value |
|---|---|---|---|---|---|---|
| | Good (393) N (%) | Poor (29) N (%) | | | | |
| Metformin | 72 (18.3) | 2 (6.9) | 0.115 | 0.47 | 8.06 | 0.08 |
| Metformin + glibenclamid | 28 (7.1) | 6 (20.7) | 0.644 | 0.19 | 0.64 | 0.66 |
| Insulin + metformin | 14 (3.6) | 2 (6.9) | 0.010 | 1.50 | | |
| Frequency of taking medication | | | | | | |
| Once a day | 29 (7.4) | 1 (3.4) | 0.438 | 0.44 | | |
| Twice a day | 364 (92.9) | 28 (96.6) | | | | |
| History of hypoglycemia | | | | | | |
| Yes | 173 (44) | 12 (41.1) | 0.782 | 0.89 | | |
| No | 220 (56) | 17 (56.2) | | | | |
| Comorbid condition | | | 0.443 | | | |
| Hypertension | 90 (22.9) | 8 (27.5) | | | | |
| Dyslipidemia | 32 (8.1) | 6 (20.6) | | | | |
| Heart failure | 19 (4.8) | 4 (13.8) | | | | |
| other | 10 (2.5) | - | | | | |
| None | 280 (71.2) | 19 (65.5) | | | | |
| Level of knowledge | | | | | | |
| Good | 312 (79.3) | 15 (51.7) | 0.001 | 0.27 | 2.87 | 0.01 |
| Poor | 81 (20.6) | 14 (48.2) | | | (1.2–6.8) | |

P value < 0.05 considered as scientifically significant

The majority (92.4%) of the current study participants had good knowledge about the complications of hypoglycemia which was higher when compared to the study conducted from south India which reported good knowledge for only one-third of the studied subjects. This might be due to the small sample size used in the latter study [29].

The majority (93.1%) of the study subjects had a good hypoglycemia prevention practice. This figure is higher than the report from south Gondar, Ethiopia (21.4%) and Tigray, Ethiopia (63.2%) [21, 26]. This difference might be due to the high accessibility of health care providers with specialty care in the current hospital.

With respect to the question "the practice on prevention of hypoglycemia while traveling", more than two-third of the current study participants (87.2%) had a good practice. This was higher than the report from south Gondar Ethiopia (36.3%) [26]. This difference can be explained from different perspectives. The time-gap between the study periods is one possible reason. There was increased access to health care facilities since the latter study had been conducted. Additionally, there were differences in the study setups, and most importantly, there was a huge knowledge difference about hypoglycemia among study subjects in the two studies.

Most of the current study subjects (81.5%) had good knowledge of self-management practices for hypoglycemia by immediate consumption of glucose (simple sugar). This is slightly higher than the study reported from Nigeria (67.6%) [30]. Forty-six percent of the study participants were diabetic for more than 5 years.

Self-monitoring of blood glucose (SMBG) was practiced by 74 (17.5%) of the current study subjects. This figure was in line with the study reported from South India (15%) and was lower than the study reported from South Africa (34%). This difference might be due to differences in health care setup and study subjects' literacy levels [29, 25]. The majority (95.5%) of participants had good practice with respect to adhering to appointment periods. Almost similar study findings were reported from south Gondar, Ethiopia (93.3%) [26].

Self-adjustment of medications was reported by more than half (57.6%) of the current study subjects which was higher compared to that of the South African (43%) and Indian (17%) studies [25, 28]. The difference may be due to the small sample size of these latter studies. The majority (93.3%) of study participants had a good practice of reporting hypoglycemic episodes to their doctors. This figure is higher compared to the study conducted in south India (48.6%) [29].

A small number (17.5%) of the current study subjects had a glucometer at their home. This was higher compared to the study report from south India (5%) and Gondar, Ethiopia (7.7%). However, it was lower than the study reported from Kwazulu-Natal, South Africa (24%), and the study report from Qatar (60.5%). This might be due to the differences in patients' economic status and their awareness regarding the importance to practice SMBG [29, 26, 31, 32].

Seventy-four (17.5%) of the current study participants reported their ability to self-monitor their glucose levels at home, even though 243 (57.6%) patients reported that they practiced self–adjustment of their medications at home. This finding suggests that some patients were self-adjusting their medication at home based on how they feel, without objectively measuring their blood sugar. This may put them at risk of developing hypoglycemia or hyperglycemia. There is a need for the provision of education about medication adjustment in DM patients [33].

In the current study, only having a good knowledge regarding hypoglycemia prevention was strongly associated with good hypoglycemia prevention practice (p = 0.01; AOR: 2.87). This means that patients with good knowledge were 2.87 times more likely to practice hypoglycemia prevention measures as compared to those patients with poor knowledge. This finding was in line with the study reported from Tigray, Ethiopia [21]. This result suggests that knowledge about hypoglycemia prevention is essential to practice hypoglycemic prevention measures.

Twenty-eight of the current study participants were hypoglycemic at the time of hospital visit which gave a prevalence of 6.6%. This was lower than different studies in the United States which reported prevalence of hypoglycemia range from 12% to 18%.This may be due to differences in the life style of the study participants between the studies. In the current study, we only used the patients' blood glucose level which they have at hand when they came for follow up. But the other studies took reported cases of hypoglycemia retrospectively [3, 31].

Even though participants had good knowledge and practice of hypoglycemia prevention strategies, a high number of participants had practice of hypoglycemia prevention without adequate knowledge of hypoglycemia prevention. This may be due to some participants may practice to prevent hypoglycemia by family support and asking some persons and even sometimes the try something by their assumption the other possible reason may be the knowledge questions were somehow it needs scientific knowledge.

## Limitation

The study design was cross-sectional. Therefore causal modeling could not be attempted. The participants were recruited from one medical center and patients who have regular visits in northwest Ethiopia. Therefore, the findings may not be generalizable to all DM patients. Future studies with a larger and more representative sample size that include patients with gestational DM and pediatric patients with DM is required.

## Conclusion

In conclusion, the practices of hypoglycemia prevention strategies among ambulatory diabetic patients were good. Knowledge of hypoglycemia prevention strategies and its practice are

essential elements for control of hypoglycemia and hypoglycemic crisis. Good knowledge of hypoglycemia prevention was strongly associated with its prevention practice.

## Recommendation

The current study showed that there are still gaps with respect to knowledge about hypoglycemia prevention and its practice. Therefore, we recommend providing education about hypoglycemia in general, its complication, its treatment, and prevention strategies when they came for follow up. We recommend that this study to be done in Ethiopia as a whole so will yield an accurate result.

## Supporting information

**S1 File.**
(DOCX)

## Acknowledgments

We would like to thank The Department of Clinical Pharmacy, School of Pharmacy, College of Medicine and Health Sciences, University of Gondar for giving us the opportunity to do these research. And most importantly we would like to thank our participants.

## Author Contributions

**Conceptualization:** Esileman Abdela Muche, Banchamlak Teferi Mekonen.

**Data curation:** Esileman Abdela Muche, Banchamlak Teferi Mekonen.

**Formal analysis:** Esileman Abdela Muche, Banchamlak Teferi Mekonen.

**Investigation:** Esileman Abdela Muche, Banchamlak Teferi Mekonen.

**Methodology:** Esileman Abdela Muche, Banchamlak Teferi Mekonen.

**Project administration:** Esileman Abdela Muche.

**Supervision:** Esileman Abdela Muche.

**Writing – original draft:** Esileman Abdela Muche, Banchamlak Teferi Mekonen.

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
