## [Decision Letter · Decision Letter 0]

11 Dec 2019

PONE-D-19-25882

Knowledge and practice of diabetic patients towards prevention of hypoglycaemia at UoGCSH:

cross-sectional study

PLOS ONE

Dear Mr Muche,

Thank you for submitting your manuscript to PLOS ONE. After careful consideration, we feel that it has merit but does not fully meet PLOS ONE’s publication criteria as it currently stands. Therefore, we invite you to submit a revised version of the manuscript that addresses the points raised during the review process.

The manuscript has multiple problems and must be completely re-written if it is to be considered. A much more focused introduction and discussion must take place. The authors are strongly encouraged to carefully read the comments of both reviewers and answer each in a revised manuscript. The authors are also encouraged to have a native English- speaking  individual fully proof the revision before it is submitted.

We would appreciate receiving your revised manuscript by Jan 25 2020 11:59PM. To enhance the reproducibility of your results, we recommend that if applicable you deposit your laboratory protocols in protocols.io, where a protocol can be assigned its own identifier (DOI) such that it can be cited independently in the future. For instructions see: http://journals.plos.org/plosone/s/submission-guidelines#loc-laboratory-protocols

We look forward to receiving your revised manuscript.

Kind regards,

Randy Wayne Bryner, Ed.D.

Academic Editor

PLOS ONE

Journal Requirements:

2. We noticed you still have some overlapping text with previous publications:http://www.ijem.in/text.asp?2013/17/5/819/117219 and

https://doi.org/10.4102/phcfm.v8i1.906, which needs to be addressed. In your revision ensure you cite all your sources (including your own works), and quote or rephrase any duplicated text outside the methods section. Further consideration is dependent on these concerns being addressed.

In the ethics statement in the Methods and online submission information, please ensure that you have specified what type of consent you obtained (for instance, written or verbal).

4. Your ethics statement must appear in the Methods section of your manuscript. If your ethics statement is written in any section besides the Methods, please move it to the Methods section and delete it from any other section. Please also ensure that your ethics statement is included in your manuscript, as the ethics section of your online submission will not be published alongside your manuscript.

5. Please include your tables as part of your main manuscript and remove the individual files. Please note that supplementary tables (should remain/ be uploaded) as separate "supporting information" files

Additional Editor Comments (if provided):

The research has the potential to be informative and add to the existing body of knowledge in this field but not in the present form as written in the submitted manuscript. A complete rewrite is necessary focusing specifically on the many good suggestions from the two reviewers. Each section (introduction, methods, results and discussion) must be refocused and rewritten. In addition, as both reviewers have stated, a minimum threshold score to consider as good knowledge or good practice must be used and justified. The authors are encouraged to collaborate with a statistician for this endeavor.

Reviewers' comments:

Reviewer's Responses to Questions

**Comments to the Author**

1. Is the manuscript technically sound, and do the data support the conclusions?

Reviewer #1: No

Reviewer #2: Partly

2. Has the statistical analysis been performed appropriately and rigorously? 

Reviewer #1: No

Reviewer #2: I Don't Know

3. Have the authors made all data underlying the findings in their manuscript fully available?

Reviewer #1: Yes

Reviewer #2: Yes

4. Is the manuscript presented in an intelligible fashion and written in standard English?

Reviewer #1: No

Reviewer #2: No

5. Review Comments to the Author

Reviewer #1: Thank you for the opportunity to review your manuscript. This study assessed knowledge and practice of preventing hypoglycemia in patients with diabetes in Gondar, Ethiopia. While the topic is interesting and should be explored, there were some limitations of the present study.

The introduction presents a nice review on hypoglycemia, but it needs to better connect to relevant literature in the area of knowledge and practice of preventing hypoglycemia. The justification for conducting this study is lacking. In the final paragraph the authors refer to previous studies – this should be expanded and these studies should be referenced. Describing the current state of understanding (from scientific literature) regarding knowledge and practice of hypoglycemia needs to be expanded in the introduction to better justify the purpose of this study in this population.

The methods – in study area and period “Diabetic patients get service two days per week” is unclear – what service is performed? Acronyms (OPD) are used within the text before spelling out their meaning. Was the questionnaire validated? As it reads now, I think there are two parts to the questionnaire – knowledge and practice – but it does not read clearly. The last sentence explains there were two categories –good and poor. The methods indicate this is based on the mean score of that section – but there’s no justification for this as the cutoff. It seems it should be more directly related to poor hypoglycemic knowledge or practice rather than based on the sampled population? Working with a statistician could allow for analysis to identify a proper cut off from the data since some individuals did present with hypoglycemia. Data collection procedures -indicates questionnaire is from previous studies but that needs to be referenced.

Results – data in the figures is also presented in the table and text. Results should be selectively presented in a single format. I would also encourage you to work with a statistician - there are several analyses that could be run with this data that might be more informative than simple descriptive statistics.

The discussion does not connect the present study to previous findings. Rather than presenting the numbers and prevalence other researchers have found, the discussion needs to connect to this work and provide an interpretation of the results. It should connect back to the original justification. Once the justification is clearly written in the introduction, connecting back to it in the discussion and using literature to support your interpretation will strengthen the manuscript. As it is now, the discussion is insufficient.

Finally, I encourage you to have this proof read by a native English speaker before resubmitting.

Reviewer #2: The authors completed a cross-sectional study at an outpatient hospital facility in Ethiopia to determine prevalence of hypoglycemia and patients’ level of knowledge about hypoglycemia and their use of various practices to prevent hypoglycemia. They discuss the results in relationship to other similar published studies and, despite a low prevalence of hypoglycemia in the study population, recommend patient education related to prevention of hypoglycemia.

This article has the potential to contribute important knowledge to the scientific literature, but not in its current form. The purpose of the study is based on the premise that this information hasn’t been collected before in this population, and that knowledge of hypoglycemia is related to better prevention of hypoglycemia. While this is a cross-sectional study and is limited in the potential to determine causality, as stated by the authors in their limitations section, the relationship between a patient’s knowledge score and their practice score could be investigated with these data (but is not).

As presented, the results of this study are difficult to interpret and do not seem to fully support the claim that the study group has good knowledge of hypoglycemia and practice on hypoglycemia prevention. The experimental methods and results are not described in sufficient detail. The authors state that the study questionnaire was used in another published study, but it is unclear as to whether the questionnaire is validated to measure what it is intended to measure. Additionally, the results need to be more detailed; specifically, the mean score for each component of the knowledge and practice score should be included in tables 3 and 4 so that the reader can understand how the whole group fares in terms of knowledge and practice of hypoglycemia prevention. This is important because the authors present the results as “good” or “poor” knowledge and practice based on how each individual score compares to the mean score of the group, but the results really don’t show whether or not the subjects have good or poor knowledge of hypoglycemia/prevention – they just show the proportion of the group that falls above the mean and the proportion that falls below the mean. If the entire sample group has limited knowledge and practice of hypoglycemia prevention (i.e. the mean is a low score out of the maximum), even scores that are less than half of the maximum could be considered “good” knowledge or practice scores. It seems that setting a minimum threshold score to consider as “good” knowledge or “good” practice would be a better way to present these results. Furthermore, it is unclear where the estimate that “72.7% of participants had good knowledge in hypoglycemia prevention” (line 1 of the discussion) came from, and similarly how the authors came to the conclusion that “93.1% of the patients had good practice” (discussion paragraph 3, line 1). These numbers are not indicated in the results in tables 3 or 4.

Finally, figures 1 and 2 are unnecessary and add no additional information beyond what is included in tables 1 and 2; they should be removed.

The discussion requires a revision for clarity - is messy and difficult to follow. Also, the authors attribute much of the differences between the findings in their study and the findings of other studies to differences in sample size. This should either be elaborated on, or more thoughtful discussion of potential reasons for disparate results should be considered. Are the authors suggesting that the other studies were not adequately powered to detect differences between their study groups? How specifically does this affect the interpretation of these results compared to other studies?

Overall, the writing is unintelligible in many places and requires significant editing to be acceptable for publication. There are numerous errors that require revision – including capitalization, punctuation, syntax, and grammatical errors – too many to type here.

Lastly, and importantly, multiple references are not cited in the manuscript, including references 1, 11, 12, 16, 17, 19, 20, 21, 28, 29, 31, and 32. Either the references should be removed from the reference list, or the manuscript should be updated to include the proper citations for these sources.

6. PLOS authors have the option to publish the peer review history of their article (what does this mean?). If published, this will include your full peer review and any attached files.

Reviewer #1: No

Reviewer #2: No

---

## [Author Response · Author response to Decision Letter 0]

15 Feb 2020

editores comment is adressed. submitted the required three document revized manuscript, revized manuscript with trackchange and response to reviwers comment.

Journal requirment was corrected as follows

1. manuscript meets PLOS ONE's style requirements, including those for file naming.

2. rephrased any duplicated text outside the methods section.overlapping text with previous publications were revized. the consent is verbal i included in the text.

3.I already included additional information regarding the survey or questionnaire used in the study. Supporting files were attached as per recommendation 

4. About ethics statment--- I removed ethics ststment form any other site except in the method section.

5. Tables are now part of manuscript

reviwers comment 

1. first question I tried to make scientifically sound with rewritten of introduction and discussion part and proof reeding

2. Has the statistical analysis been performed appropriately and rigorously? Now i made analysis to find out associated factors with binary logestic regression analysis. Finally new table was made and appropraitely interpreted.

3. Is the manuscript presented in an intelligible fashion and written in standard English? I completely proff read and re written to make it easy to understand.

reviwers comment was adressed and attached as per request.

---

## [Decision Letter · Decision Letter 1]

26 Mar 2020

PONE-D-19-25882R1

Hypoglycaemia prevention practice and its associated factors among diabetes patients at university teaching hospital in Ethiopia: cross-sectional study

PLOS ONE

Dear Mr Muche,

Thank you for submitting your manuscript to PLOS ONE. After careful consideration, we feel that it has merit but does not fully meet PLOS ONE’s publication criteria as it currently stands. Therefore, we invite you to submit a revised version of the manuscript that addresses the points raised during the review process.

We thank the authors for their revised manuscript but there are still some issues that must be addressed. Please answer each of the comments made in the review. Additionally, the manuscript still needs to be thoroughly reviewed by a native English speaking person.

We would appreciate receiving your revised manuscript by May 10 2020 11:59PM. To enhance the reproducibility of your results, we recommend that if applicable you deposit your laboratory protocols in protocols.io, where a protocol can be assigned its own identifier (DOI) such that it can be cited independently in the future. For instructions see: http://journals.plos.org/plosone/s/submission-guidelines#loc-laboratory-protocols

We look forward to receiving your revised manuscript.

Kind regards,

Randy Wayne Bryner, Ed.D.

Academic Editor

PLOS ONE

Reviewers' comments:

Reviewer's Responses to Questions

**Comments to the Author**

1. If the authors have adequately addressed your comments raised in a previous round of review and you feel that this manuscript is now acceptable for publication, you may indicate that here to bypass the “Comments to the Author” section, enter your conflict of interest statement in the “Confidential to Editor” section, and submit your "Accept" recommendation.

Reviewer #1: (No Response)

2. Is the manuscript technically sound, and do the data support the conclusions?

Reviewer #1: No

3. Has the statistical analysis been performed appropriately and rigorously? 

Reviewer #1: Yes

4. Have the authors made all data underlying the findings in their manuscript fully available?

Reviewer #1: Yes

5. Is the manuscript presented in an intelligible fashion and written in standard English?

Reviewer #1: No

6. Review Comments to the Author

Reviewer #1: Thank you for revising your manuscript. The introduction still does not fully justify this study - if you're running multivariate regressions it indicates you're trying to find predictors - that needs to be included in the introduction. In the current version the operational definitions are still unclear. You state that Poor knowledge or practice is when patients scored below 50% of the mean score. That implies that 50% of the population will have "good" and the other half will have "poor" knowledge or practice. The threshold for good and bad should not be based on the population - if the entire sample scores <25% correct that would suggest nobody had good practice or knowledge, but according to your operational definition the top 50% would still be classified as "good". However, when reading the results it seems this is not, in fact, how you are using "good" or "poor" since 77.% had good knowledge of hypoglycemia prevention. Please clarify your operational definition. The units of birr will not make sense to readers outside of the study area - perhaps you could stratify based on the average income of the area. The limitation section acknowledges that these findings are limited to the region under study, but it's also limited to patients that regularly visit the hospital. There are likely a great number of individuals with diabetes that do not visit the clinic regularly and are therefore less likely to be monitoring blood sugar. This manuscript still needs to be revised and proofread by a native English speaker.

7. PLOS authors have the option to publish the peer review history of their article (what does this mean?). If published, this will include your full peer review and any attached files.

Reviewer #1: No

---

## [Author Response · Author response to Decision Letter 1]

4 May 2020

Response to reviewer

1. Introduction should include predictors to hypoglycemia prevention practice.

As far as investigators knowledge there is no literatures which showed predictors of hypoglycemia prevention practice except reference 21 and 26. And these reference are reviewed under paragraph 8 under introduction.

2. Operational definition still unclear?

Now it becomes clarified

Good Knowledge: when patients scored > 6 on the knowledge assessment questions.

Poor Knowledge: when patients scored < 6 on the knowledge assessment questions.

Good Practice: when patients scored >8.5 on the practice assessment questions.

Poor practice: when patients scored < 8.5 on the practice assessment questions.

3. Economic status should be stratified based on the average income and change to international currency.

This stratification is based on the study area salary scale. But we changed with USD by considering current currency exchange.

4. These findings are limited to the region under study, but it's also limited to patients that regularly visit the hospital.

Accepted and stated to limitation part this study.

---

## [Decision Letter · Decision Letter 2]

27 May 2020

PONE-D-19-25882R2

Hypoglycaemia prevention practice and its associated factors among diabetes patients at university teaching hospital in Ethiopia: cross-sectional study

PLOS ONE

Dear Dr. Muche,

Thank you for submitting your manuscript to PLOS ONE. After careful consideration, we feel that it has merit but does not fully meet PLOS ONE’s publication criteria as it currently stands. Therefore, we invite you to submit a revised version of the manuscript that addresses the points raised during the review process.

In particular, there are several typing and grammatical errors, etc. that detract from the scientific merit of this work. For the manuscript to be consider it is imperative that it be thoroughly reviewed and edited by a native English speaker.

We look forward to receiving your revised manuscript.

Kind regards,

Randy Wayne Bryner, Ed.D.

Academic Editor

PLOS ONE

Reviewers' comments:

Reviewer's Responses to Questions

**Comments to the Author**

1. If the authors have adequately addressed your comments raised in a previous round of review and you feel that this manuscript is now acceptable for publication, you may indicate that here to bypass the “Comments to the Author” section, enter your conflict of interest statement in the “Confidential to Editor” section, and submit your "Accept" recommendation.

Reviewer #1: All comments have been addressed

2. Is the manuscript technically sound, and do the data support the conclusions?

Reviewer #1: Yes

3. Has the statistical analysis been performed appropriately and rigorously? 

Reviewer #1: Yes

4. Have the authors made all data underlying the findings in their manuscript fully available?

Reviewer #1: Yes

5. Is the manuscript presented in an intelligible fashion and written in standard English?

Reviewer #1: No

6. Review Comments to the Author

Reviewer #1: Thank you for addressing and sufficiently responding to the requested revisions. Other than a careful review and edit by a native English speaker I have no other comments.

7. PLOS authors have the option to publish the peer review history of their article (what does this mean?). If published, this will include your full peer review and any attached files.

Reviewer #1: No

---

## [Author Response · Author response to Decision Letter 2]

7 Jun 2020

Reviewer #1: Thank you for addressing and sufficiently responding to the requested revisions. Other than a careful review and edit by a native English speaker I have no other comments.

As stated above Ousman Abubeker who is fluent in English language thoroughly reviewed and now I revised according to his comment.

---

## [Editor Report · Decision Letter 3]

12 Jun 2020

PONE-D-19-25882R3

Hypoglycaemia prevention practice and its associated factors among diabetes patients at university teaching hospital in Ethiopia: cross-sectional study

PLOS ONE

Dear Dr. Muche,

Thank you for submitting your manuscript to PLOS ONE. After careful consideration, we feel that it has merit but does not fully meet PLOS ONE’s publication criteria as it currently stands. Therefore, we invite you to submit a revised version of the manuscript that addresses the points raised during the review process.

It appears that the authors are ignoring the reviewers comments to have the manuscript carefully edited by a native English speaking individual. There are still several grammatical mistakes that need to be corrected before it can be considered for publication.

We look forward to receiving your revised manuscript.

Kind regards,

Randy Wayne Bryner, Ed.D.

Academic Editor

PLOS ONE

---

## [Author Response · Author response to Decision Letter 3]

26 Jun 2020

manuscript carefully edited by a native English speaking individual. I did it as I mentioned it above.

no more comment from reviewer.

---

## [Editor Report · Decision Letter 4]

11 Aug 2020

Hypoglycemia prevention practice and its associated factors among diabetes patients

at university teaching hospital in Ethiopia: cross-sectional study

PONE-D-19-25882R4

Dear Dr. Muche,

We’re pleased to inform you that your manuscript has been judged scientifically suitable for publication and will be formally accepted for publication once it meets all outstanding technical requirements.

Kind regards,

Randy Wayne Bryner, Ed.D.

Academic Editor

PLOS ONE

Additional Editor Comments (optional): Thank you for your careful review and corrections of the grammatical concerns and English issues. 
---

## [Editor Report · Acceptance letter]

13 Aug 2020

PONE-D-19-25882R4 

Hypoglycemia prevention practice and its associated factors among diabetes patients
at university teaching hospital in Ethiopia: cross-sectional study 

Dear Dr. Muche:

I'm pleased to inform you that your manuscript has been deemed suitable for publication in PLOS ONE. Congratulations! Your manuscript is now with our production department. 

Kind regards, 

on behalf of

Dr. Randy Wayne Bryner 

Academic Editor

PLOS ONE